# Tubulocystic Renal Cell Carcinoma Is Not an Indolent Tumor: A Case Report of Recurrences in the Retroperitoneum and Contralateral Kidney

**DOI:** 10.3390/medicina57080851

**Published:** 2021-08-21

**Authors:** Tae-Soo Choi, Dong-Gi Lee, Kyu-Yeoun Won, Gyeong-Eun Min

**Affiliations:** 1Department of Urology, Kyung Hee University College of Medicine, Seoul 05278, Korea; taesoochoi85@hanmail.net (T.-S.C.); urology@khu.ac.kr (D.-G.L.); 2Department of Pathology, Kyung Hee University College of Medicine, Seoul 05278, Korea; wonki96@khu.ac.kr

**Keywords:** carcinoma, renal cell, recurrence, neoplasm metastasis

## Abstract

Tubulocystic renal cell carcinoma (RCC) is a rare subtype of RCC that was recently included in the 2016 World Health Organization classification of tumors of the kidney. Most of these tumors exhibit indolent behavior with low metastatic potential. However, here we report a case of recurrent tubulocystic RCC with aggressive features in the retroperitoneum and contralateral kidney treated with targeted agents and radiofrequency ablation.

## 1. Introduction

Tubulocystic renal cell carcinoma (RCC) is a subtype of RCC only recently included in the 2016 World Health Organization (WHO) classification of tumors of the kidney [1]. Most cases have low metastatic potential. Some tubulocystic RCC cases are misdiagnosed as renal cysts [2]. To date, the treatment of metastatic tubulocystic RCC has not been established. Herein, we report a case of recurrent tubulocystic RCC with aggressive features in the retroperitoneum and contralateral kidney treated with targeted agents and radiofrequency ablation (RFA).

## 2. Case Report

A 60-year-old man was referred to our department for a huge left renal cyst identified on computed tomography (CT), which revealed an approximately 14-cm-sized renal cyst with thin septa at the upper pole of the left kidney was observed (Figure 1). The preoperative CT findings were consistent with Bosniak classification II renal cyst. Subsequently, we performed laparoscopic renal cyst marsupialization. On pathologic examination, the tumor comprised a mixture of variably sized cysts and tubules, which were lined by a single layer of flattened, cuboidal or columnar cells. Hobnailing was observed. The subepithelial area demonstrated fibrous stroma and poorly differentiated tumor cells (Figure 2 and Figure 3a). The pathological diagnosis was tubulocystic RCC. As the surgical margin status could not be evaluated, a radical nephrectomy was chosen. Two weeks after renal cyst marsupialization, the patient underwent radical nephrectomy. Pathologic examination revealed tumor cells in the residual kidney. Poorly differentiated foci of 1.5 × 1.5 cm with pleomorphic nuclei and prominent nucleoli were observed (Figure 3b). The tumor cells were positive for cytokeratin, vimentin, and AMACR (Figure 3c); and negative for cytokeratin 7, and CD10. After 18 months, multiple small hypodense lesions in the left subphrenic space and along the lateral portion of the left retroperitoneal space were noted on an abdominal CT scan. The lesion density was <10 HU, and the masses were cystic (Figure 4). The patient received a weekly dose of 25 mg temsirolimus intravenously. After eight cycles of temsirolimus, the appearance of multiple retroperitoneal nodules was reduced on the abdominal CT. Weekly temsirolimus treatment was continued. After an additional eight cycles of temsirolimus, recurrent RCC was detected in the contralateral kidney (Figure 5). RFA of the recurrent RCC in the right kidney was performed after confirming the pathology through a percutaneous renal mass biopsy. Treatment of the retroperitoneal nodules was continued with 50 mg of sunitinib. At the 9-month follow-up, the lesions were stable disease according to the RECIST criteria. The patient continually took sunitinib 50 mg without major side effects other than mild fatigue and subclinical hypothyroidism, and his renal function was maintained at CKD stage 3a.

## 3. Discussion

Tubulocystic RCC is a rare subtype of RCC that was recently included in the 2016 WHO classification of kidney [1]. The majority of these tumors exhibit indolent behavior with low metastatic potential. Only a few cases of tumors with local recurrence and metastases to the lymph node, liver, bone, pleura, and peritoneum have been reported [3,4,5]. The pre-operative diagnosis of tubulocystic RCC is challenging. If there is a solid portion inside the renal mass, the possibility of RCC can be considered, but if it has only a cystic component, these tumors are difficult to distinguish from tubulocystic RCC or renal cysts. In Cornelis’s study of 16 tubulocystic RCC studied by CT, two were considered solid, seven were cystic, and seven were indeterminate [6]. Among cystic lesions, one was classified as purely cystic (Bosniak I), one as Bosniak II, one as Bosniak IIF, and four as Bosniak IV. In our case, the preoperative diagnosis was Bosniak II renal cyst based on CT findings. Detecting contrast enhancement on CT is critical and remains a challenge due to the very low vascularity of tubulocystic RCC and the small number of solid tissue components [6]. Magnetic resonance imaging is very useful for demonstrating the microcystic nature of these tumors, owing to its superior contrast resolution [7,8]. In addition, the ultrasound (US) is useful in identifying the tubulocystic RCC [9]. The US pattern of tubulocystic RCC exhibits high echogenicity and posterior acoustic enhancement because of its multicystic characteristics separated by multiple thin septae [6].

Tubulocystic RCC is a dominantly cystic renal epithelial neoplasm. Macroscopically, it comprises multiple small-to-medium-sized cysts and has a spongy cut surface. The nuclei were enlarged according to WHO/International Society of Urological Pathology (ISUP) grade 3 nucleoli. The cytoplasm had abundant eosinophilic and oncocytoma-like features. Tumors presented grossly as a complex cystic mass, characteristic in male patients (M:F ratio of 7:1) during their seventh decade [4]. These tumors demonstrated a consistent morphology of variably cystically dilated tubules, admixed with a background of fibrous stroma, and lined by markedly atypical cells with eosinophilic cytoplasm and high-grade nuclei with prominent nucleoli (ISUP nucleolar grade 3) [4]. Most reported tumors have been at a low stage, with only rare reports of clinical progression and aggressive behavior [10,11,12]. The coexistence of poorly differentiated foci indicates a worse prognosis than that of the common type of tubulocystic RCC [13,14]. Al-Hussain and Zhao reported several cases of tubulocystic RCC with poorly differentiated foci that had metastases and local recurrences in the abdomen, pelvic cavity and bones [13,15]. Tubulocystic RCC with poorly differentiated foci should be distinguished from the hereditary leiomyomatosis renal cell carcinoma syndrome (HLRCC). Most tubulocystic RCCs with poorly differentiated foci have a characteristic nucleus in the form of a large nucleus with prominent inclusion-like eosinophilic nucleoli reminiscent of nuclear features described in HLRCC [16]. For a proper differential diagnosis, a thorough investigation of family history and genetic testing should be considered.

Complete surgical excision is the principal treatment modality for RCC. However, tubulocystic RCC is often misdiagnosed as a renal cyst, and some tubulocystic RCC is treated with renal cyst marsupialization, as was in our case [2]. In this case, risk of local recurrence even after radical nephrectomy following renal cyst marsupialization could be pre-sent. To date, the treatment for metastatic tubulocystic RCC has not been established. Treatment must be individualized and, therefore requires a multidisciplinary approach. There have been several case reports on the administration of chemotherapeutic agents or targeted agents. A few case reports have suggested a partial response to sunitinib (a tyro-sine kinase inhibitor) and everolimus (a mammalian target of rapamycin (mTOR) inhibitor) [17,18,19]. In our case, the patient achieved a partial response after the initial administration of temsirolimus, including a decrease in size and disappearance of several nodules. However, in the subsequent response evaluation, the effect of treatment was not sustained, and recurrence was observed in the contralateral kidney. For locally recurrent or oligometastatic RCC, metastasectomy, stereotactic body radiation therapy, or ablative treatment should be considered according to the National Comprehensive Cancer Network guidelines [20]. RFA is an appropriate treatment option for small RCC with durable onco-logical outcomes of 94–96.1% disease-free survival and low complication rates [21,22]. Integrated with a systemic treatment strategy, RFA is safe and effective for the treatment of metastatic disease from RCC with good overall survival and long systemic treatment-free survival [23]. Concomitant RFA of recurrent RCC and targeted agents for metastatic lesions is a feasible approach in this challenging scenario.

In conclusion, most tubulocystic RCC have features of indolent tumors and rarely recur. However, given that tubulocystic RCC with poorly differentiated foci has an aggressive clinical course, more detailed follow-up is required. In cases of local and distant metastases, a multimodal treatment strategy is required.

## Figures and Tables

**Figure 1 medicina-57-00851-f001:**
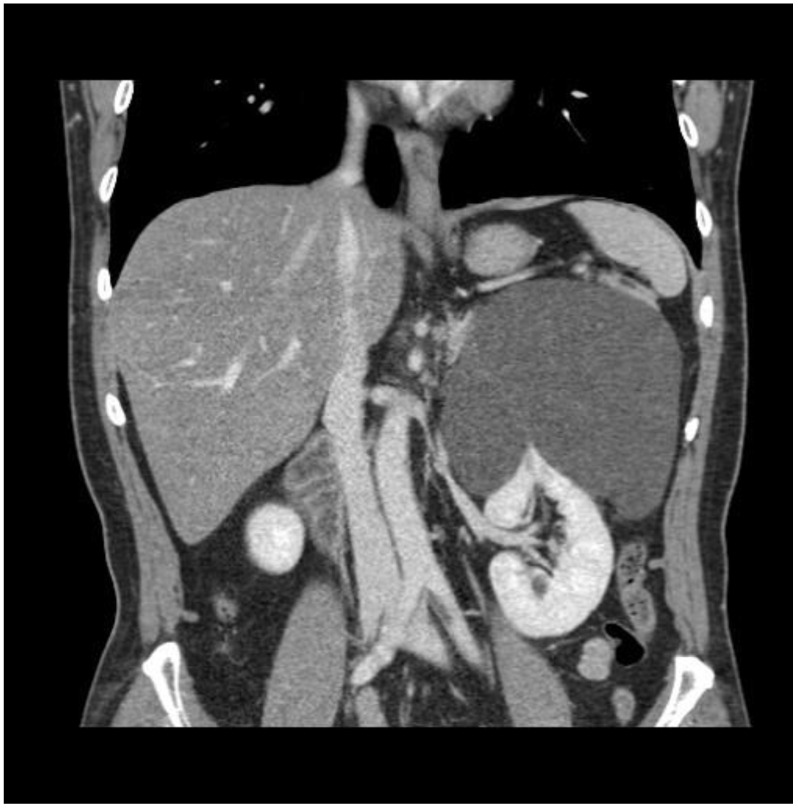
Preoperative CT scan revealed a 14 cm-sized renal cyst with thin septa at the upper pole of left kidney.

**Figure 2 medicina-57-00851-f002:**
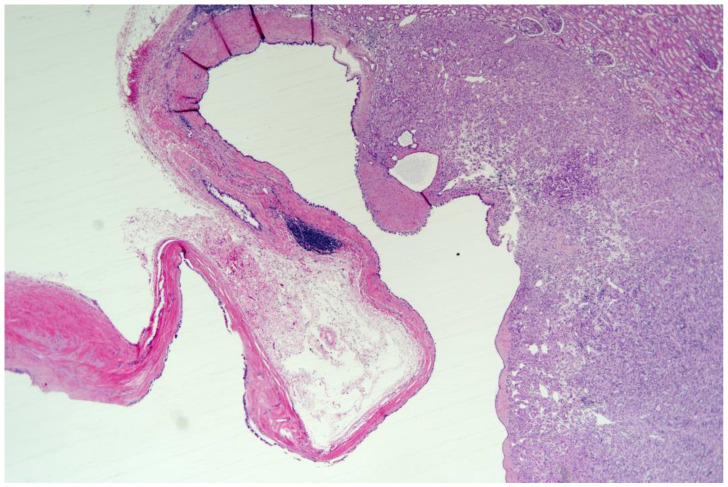
The tumor comprised a mixture of variably sized cysts and tubules (left side). The right side showed poorly differentiated tumors. (H&E stain, ×20).

**Figure 3 medicina-57-00851-f003:**
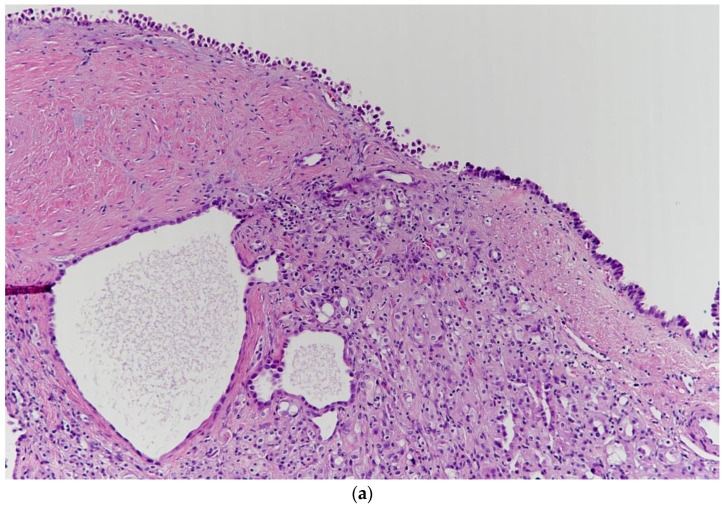
(**a**) The tubules and cysts were lined by a single layer of flattened, cuboidal or columnar cells. Hobnailing was present. The subepithelial area showed fibrous stroma and poorly differentiated tumor cells (H&E stain, ×100). (**b**) The tumor contained poorly differentiated cells that had pleomorphic nuclei and prominent nucleoli (H&E stain, ×400). (**c**) The poorly differentiated area showed positivity for AMACR (×200).

**Figure 4 medicina-57-00851-f004:**
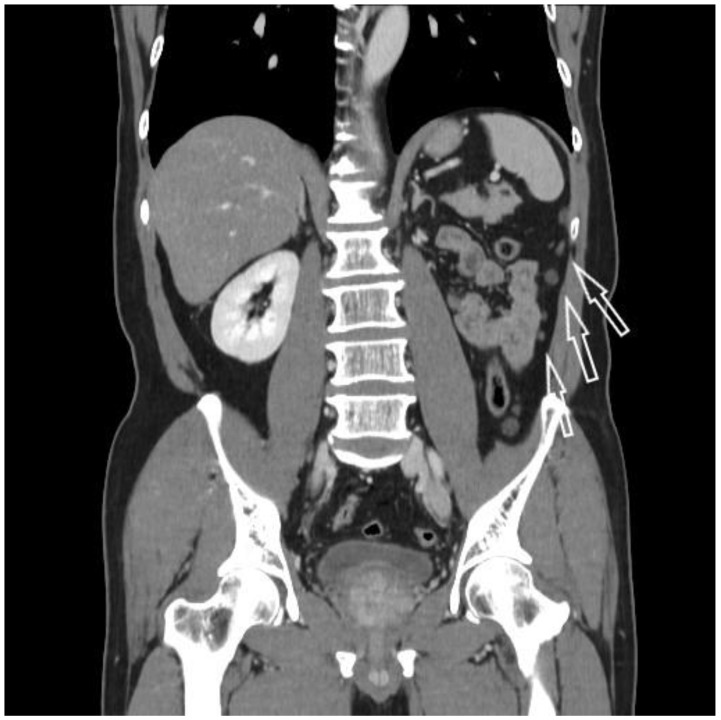
Eighteen months later, multiple small hypodense lesions in the left subphrenic space and along the lateral portion of the left retroperitoneal space (arrows) were noted on an abdominal CT scan.

**Figure 5 medicina-57-00851-f005:**
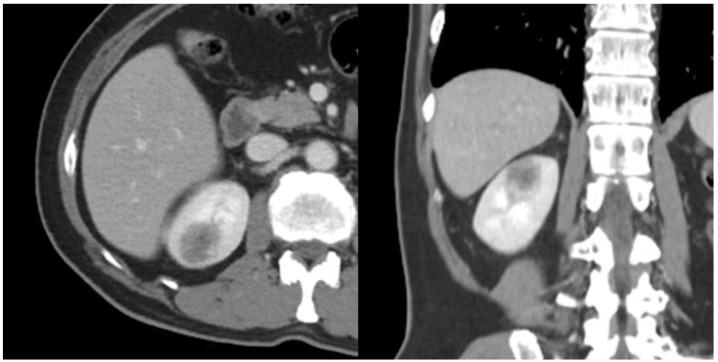
A hypodense lesion of 2.0 cm was observed in the upper pole of the right kidney.

## Data Availability

Data sharing not applicable.

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
