# Peer review of "Tubulocystic Renal Cell Carcinoma Is Not an Indolent Tumor: A Case Report of Recurrences in the Retroperitoneum and Contralateral Kidney"

_medicina, 2021, doi:10.3390/medicina57080851_

Round 1

Reviewer 1 Report

Article entitled "Tubulocystic Renal Cell Carcinoma Is Not an Indolent Tumor: A Case Report of Recurrences in Retroperitoneum and Contra- lateral Kidney" is reporting tubulocystic RCC (TC-RCC) as rare variant of RCC which usually shows indolent course, and in this case is recidivant tumor with  true malignant features.

I have few major remarks on the article:

At the Figures 2 and 3 where histology is shown there is no classical picture of TC-RCC histology which should be composed of tubular and cystic spaces separated by fibrous stroma. Cysts are usually lined by a single layer of flattened, cuboidal or columnar cells; hobnailing may be present with modest eosinophilic cytoplasm, it may resemble oncocytoma cells. Nuclei are uniform, round with ISUP grade 3 nucleoli. It is not shown at the Figures and not even described in the text as pathohistological report so Im not sure in the diagnosis. Please provide report and Figures. Also Figures are showing areas presenting poorly differentiated parts of the tumor which are in the article described as clear cell like or collecting duct like areas and it is not visible. Also, there are more reliable immunohistochemistry markers than vimentin. CK7, AMACR and PAX-8 are usually positive in TC-RCC and CAIX is negative but should be positive in clear cell RCC if present. There is also no description of immunohistochemistry findings from the pathologist report in the article. 

There is note about recurrent RCC in the contralateral kidney which was confirmed by biopsy and it should be considered second primary or metastatic RCC.

In the conclusion, data about patient and his PHD report is not shown in detail. 

In the discussion part there are no data about pathohistological variety of TC-RCC and also authors should mention Hereditary Leiomyomatosis and Renal Cell Carcinoma (HLRCC) syndrome which can show TC-RCC- like areas. It is high grade tumor with solid, papillary and glandular architecture but may have cystic or TC-RCC areas. Characteristic are large nucleoli with perinuclear halos. There is also FH gene mutation. It is very aggressive form of tumor and the authors didnt mention that they consider it in their differential-diagnosis. 

This article requires extensive changes before publication.

Author Response

I really appreciated for your review and comments on our manuscript.

As you pointed out, the figures have been changed, and related contents have been added in the  text.

Also, we described the patient's status at the last follow-up to the case report section.

In the discussion section, we have supplemented the HLRCC-related content pointed out by reviewers.

Reviewer 2 Report

This report describes the case of a recurrent tubulocystic RCC with aggressive features in the retroperitoneum and contralateral kidney. The paper is well written and the case is interesting as it presents an aggressive variant of TcRCC, which is usually more indolent. 

In my opinion, several modifications could be made to improve the quality of the work:

1) You correctly state that these tumors can be confounded with renal cysts: you could discuss more thoroughly the diagnostic workup that might avoid the misdiagnosis, including the role of ultrasound which easily distinguish between solid and liquid features. You could cite a work of Oderda et al, Minerva Urol Nefrol 2016

2) I would better describe the patient status at the end of the treatment: from what I have understood he is not progressing but he is not disease free, either. Is it correct? What are his general conditions? Side effects of the treatments? His renal function?

3) Minor synthax and grammar corrections are needed

Author Response

I am very grateful for your review and comments on my manuscript.

  • You made a very important point and comment where I was overlooking.

I fully agree with your opinion that ultrasound provides important information for differential diagnosis of tubulocystic RCC from renal cyst. I will revise and supplement the points of my manuscript by citing the article you recommended.

  • In the last examination, the patient's lesions were in a stable disease condition according to the RECIST criteria. He is taking the sunitinib 50mg without major side effects other than mild fatigue and subclinical hypothyroidism, and his renal function is maintained at CKD stage 3a. So, we described the patient's status at the last follow-up to the case report section.
  • Through the English proofreading service, we corrected the grammar and syntax.

Round 2

Reviewer 1 Report

After I have reed new version of the manuscript I agree with changes authors implemented and in the current form it is acceptable for publication.